# Bioindication of the Influence of Oil Production on Sphagnum Bogs in the Khanty-Mansiysk Autonomous Okrug–Yugra, Russia

**Olga Skorobogatova** [1],[*], **Elvira Yumagulova** [1], **Tatiana Storchak** [1] **and Sophia Barinova** [2]

[1] Department of Ecology, Nizhnevartovsk State University, 56, Lenin Street, Nizhnevartovsk, Khanty-Mansiysk Autonomous Okrug–Yugra 628605, Russia; elvirau2009@yandex.ru (E.Y.); tatyanastorchak@yandex.ru (T.S.)

[2] Institute of Evolution, University of Haifa, Abba Khoushi Ave, 199, Mount Carmel, Haifa 3498838, Israel; sophia@evo.haifa.ac.il

[*] Correspondence: olnics@yandex.ru; Tel.: +7-912-937-0861

**Abstract:** Algal diversity in the bogs of the Ershov oil field of the Khanty-Mansiysk Autonomous Okrug–Yugra (KMAO-Yugra) with the gradient of oil pollution between 255 and 16,893 mg kg$^{-1}$ has been studied with the help of bioindication methods and ecological mapping. Altogether 91 species, varieties, and forms of algae and cyanobacteria from seven divisions have been revealed for the first time from seven studied sites on the bogs. Charophyta algae prevail followed by diatoms, cyanobacteria, and euglenoids. The species richness and abundance of algae were maximal at the control site, with charophytic algae prevailing. The species richness of diatoms decreased in the contaminated area, but cyanobacteria were tolerated in a pH which varied between 4.0 and 5.4. Euglenoid algae survived under the influence of oil and organic pollution. Bioindication revealed a salinity influence in the oil-contaminated sites. A comparative floristic analysis shows a similarity in communities at sites surrounding the contaminated area, the ecosystems of which have a long-term rehabilitation period. The percent of unique species was maximal in the control site. Bioindication results were implemented for the first time in assessing the oil-polluted bogs and can be recommended as a method to obtain scientific results visualization for decision-makers and for future pollution monitoring.

**Keywords:** freshwater algae; ecology; bioindication; oil pollution; sphagnum bogs; wetlands; Russia

---

## 1. Introduction

Oil and products released during extraction from both marine and freshwater habitats are xenobiotics for biota [1]. Oil spills are mostly discussed for the coastal zone of the ocean rather than for freshwater bodies. Nevertheless, oil spills on freshwater bodies are more frequent and often more destructive to the environment [2]. Whereas low-latitude oil production areas have received much attention in research and regulation, the northern areas have been studied for the oil impact on ecosystems because their aquatic organisms are more sensitive to impact [3]. All types of freshwater organisms are susceptible to the deadly effects of oil spills, including mammals, aquatic birds, fish, insects, micro-organisms, and vegetation. Moreover, the effects of oil spills on freshwater micro-organisms, invertebrates, and algae tend to move up the food chain and affect other species also [1]. Standing water, such as in marshes or swamps and bogs with little water movement, is likely to incur more severe impacts than flowing water because spilled oil can remain in standing water for a long time. In standing water habitats, the affected ecosystem may take many years to be restored. Oil spills have a wide impact on many interrelated species, including wetland vegetation, which provide

habitats for many living organisms [1]. Spilled oil can cover bog surfaces killing mosses and disrupting wetland ecosystems.

The northern Siberia region of Russia is full of fields for oil production, and therefore can be an important object of many researches. Whereas assessment of the oil production influence on different types of organisms has been started in this region, the influence on algae as the first level of the aquatic ecosystem has never been studied, up to now [4].

The Khanty-Mansiysk Autonomous Okrug–Yugra (hereinafter KMAO-Yugra) occupies a vast territory in western Siberia. According to hydrological-climatic zoning, the territory of the KMAO-Yugra represents the zones of excessive and very excessive moisture with insufficient heat supply. The swampiest of the subzone of the middle taiga of the region is 40%, with a prevalence of high-type sphagnum bogs [5]. The main branches of the KMAO-Yugra are used in geological exploration for oil and gas, in oil and gas refining industries, construction of roads, industrial and civil facilities, logging, fishing, and hunting. In particular, deep electric centrifugal pumps carry out oil production at the Ershov field. Here, oil and its refined products are found in the environment when drilling and gushing from exploratory wells, in cases of vehicle accidents, in oil and product pipelines, in leakage of columns in wells, and in processing equipment when discharging untreated field wastewater. The largest spills are observed in areas with a general level of relief and a high level of groundwater (high-oligotrophic bogs). A study of the high wetlands in the KMAO-Yugra shows that a strong leaching regime is typical [6], and that the horizontal transport of pollutants [7] as well as the mosaic nature of their distribution has also been found [8].

Algae are good indicators of the intensity and quality of pollution because they show an integrated response to the impact, even if it occurred in previous seasons. Impact assessment can be carried out not only according to the ecological preferences of indicator species that make up the affected communities, but also according to their taxonomic composition [9,10].

Diversity in algae of sphagnum bogs of the KMAO-Yugra was studied sporadically [8,11–14], with only the information on the effect of oil field exploitation on the diatom community presented [15].

This study aimed to reveal the algae diversity in the bogs of the Ershov oil field of the KMAO-Yugra and to assess the oil pollution impact on the bog ecosystems using methods of algal bioindication and statistical mapping.

## 2. Study Site Description

Studies were conducted on the territory of the KMAO-Yugra oil production area, in the floodplain of the Vakh River, in the Vakhovsky woodland area of the first floodplain terrace [7]. The surface water of the Vakh River basin of the studied areas is low alkaline with low dissolved solids concentration but acidity increased in the bogs.

Seven experimental plots were selected in the bogs of the Ershov oil fields, which were referenced or contaminated with oil products (Figure 1). The produced water was formed as a result of the separation of the oil emulsion. The timing and implementation of restoration activities were taken into account. The referenced site is a sphagnum-cranberry bog (site 1, Figure 2a) not exposed to direct oil pollution. Along the outskirts, it passes into a small-leaved forest. On site 2, an oil spill occurred 20 years ago, at the same time work was done on the restoration (rehabilitation). On site 3, an oil spill occurred ten years ago, and four years later work was done on the restoration. Site 4 was polluted seven years ago. Restoration work has not been carried out, the site is self-healing, as well as site 5, which was polluted ten years ago. Site 6, contaminated 20 years ago, is still self-healing; only plowing was done. Site 7 is a sedge-sphagnum bog contaminated with oil products ten years ago, which has not been rehabilitated and has a high content of oil products (Table 1, Figure 2b).

The studied sites differed in water cut and density of the projective cover of higher plants.

**Table 1.** Averaged chemical variables in the studied sites of the sphagnum bogs of the Ershov oil field in the Vakh River with geographical coordinates, year of impact, and rehabilitation state.

| Site | 1 | 2 | 3 | 4 | 5 | 6 | 7 |
|---|---|---|---|---|---|---|---|
| North | 77° 44′ 4.5132″ | 77° 48′ 11.9088″ | 77° 44′ 41.2908″ | 77° 45′ 5.3712″ | 77° 47′ 14.514″ | 77° 45′ 59.9184″ | 77° 45′ 26.1936″ |
| East | 61° 15′ 3.7764″ | 61° 10′ 24.5568″ | 61° 14′ 55.41″ | 61° 10′ 8.7672″ | 61° 11′ 52.3572″ | 61° 11′ 56.3568″ | 61° 10′ 15.8124″ |
| pH | 4.1–4.8 | 5.4 | 4.0–4.5 | 4.6 | 4.3 | 4.2 | 4.1 |
| Mineral oil, mg kg$^{-1}$ | 255.65 ± 1.50 | 419.40 ± 1.30 | 1203.10 ± 25.00 | 1500.38 ± 21.20 | 1715.47 ± 4.56 | 3308.52 ± 5.33 | 16,893.80 ± 2.54 |
| T °C | 21 | 21 | 22 | 22 | 22 | 23 | 22 |
| Years after pollution | 0 | 20 | 10 | 7 | 10 | 20 | 10 |
| State | Unimpaired | Reclaimed | Reclaimed | Self-healing | Self-healing | - | Self-healing |

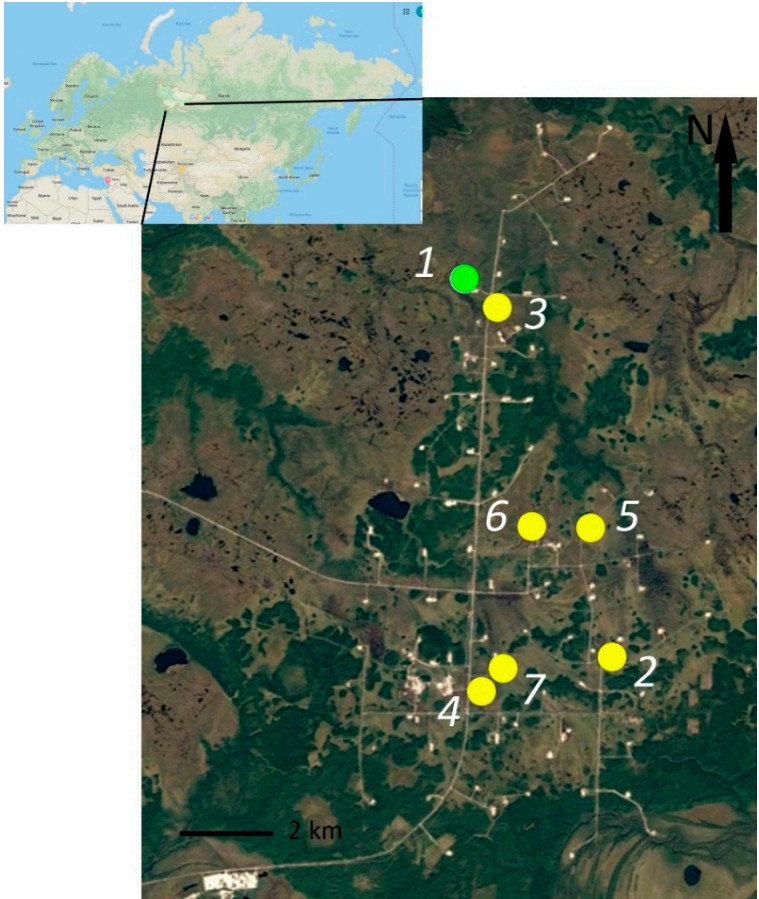

**Figure 1.** Study sites in the sphagnum bogs of the Ershov oil field in the Vakh River basin: number of studied sites, 1–7; green point, unpolluted site 1; yellow points, polluted sites 2–7.

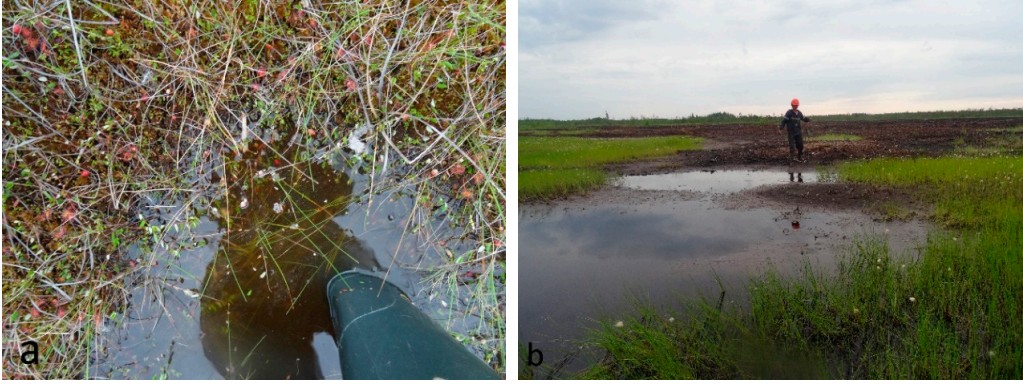

**Figure 2.** Study sites in the sphagnum bogs of the Ershov oil field in the Vakh River basin: site 1, referenced (**a**); site 7, polluted by mineral oil (**b**).

## 3. Material and Methods

Field studies for the initial assessment of impact screening were carried out mid-July 2018 at seven sites of the Ershov oil field, in the floodplain of the estuary of the Vakh River, in the Vakhovsky woodland area of the first floodplain terrace [7]. Sampling was carried out during the summer season because the impact on the aquatic ecosystem of the bogs is maximal in the ice-free period when the bog water level is maximal. The development of algal communities is highest in summer.

The material for the work included 34 samples of phytoplankton. The collection, sample preparation, and study of algae were carried out according to the methods adopted in algology [16].

Water samples with a volume of 1 liter were taken by squeezing the alloy, followed by fixation in a 4% solution of neutral formaldehyde and processing by the sediment gravimetric method. Counting cells of each species was carried out in the Nageott chamber. In parallel with the collection of algological material, the temperature and active reaction of water were measured with the Waterproof Pocket pH Tester EcoTestr pH 2 from Oakton Instruments. The nitric-nitrogen concentration in 0.5 L$^{-1}$ water samples from the bog was measured sporadically in different stations and in different years with the photometric method using sodium salicylate in the defining range of concentrations from 0.1 to 2.0 mg L$^{-1}$ [17].

The mineral oil concentration was assessed in the Nizhnevartovsk State University chemical laboratory. Samples of the swamp mass (soil) were taken using the envelope method from a depth of 0–25 cm. The size of the test site was $10 \times 10$ m. A combined sample was made, weighing 1 kg, by mixing five-point samples of 200 g each, which was placed in a plastic bag and numbered. Soil samples were taken in bags made of polyethylene.

Samples of the swamp mass (soil) were dried at room temperature to an air-dry state. Then, mechanical inclusions (undecomposed roots, plant residues, stones, etc.) were removed, crushed using a laboratory homogenizer, and rubbed through a sieve with a mesh diameter of 0.5 mm. A sample of bog mass (soil) weighing 100 g was taken from the sample, which was air dried to constant weight. For analysis, two parallel samples of 5 g each were used. The content of oil products was determined by an IR spectrometry on analyzers of oil products. The method is consistent with the extraction of oil products from the swamp mass (soil) and bottom sediments with carbon tetrachloride, the chromatographic separation of oil products from related organic compounds of other classes, and the quantitative determination of oil products by the absorption intensity in the infrared region of the spectrum.

Diatoms were determined in permanent preparations made by enclosing their flaps in Canadian balsam. The liberation of the shells of diatoms from the organic substances was carried out by the method of cold burning. Microscopy was performed using light microscopes Nikon ECLIPSE E200 and OLYMPUS SX4, with a magnification of ×100–×2000. To identify the algae taxa, domestic and foreign handbooks were used; changes in the modern nomenclature of algae were taken into account [18]. Environmental preferences of algae species, bioindication assessment methods, and ecological mapping are taken from [10,19,20]. Statistical data processing was performed in the BioDiversity Pro, the CANOCO [21], Statistica 12.0, and JASP (Jeffreys's Amazing Statistics Program) of R-statistics [22] programs.

## 4. Results

The studied sites differ in different water temperatures during the collection of algae (from 21 to 23 °C). The activity of the hydrogen ion ranged from 4.0 to 5.4. The color of the water ranged from yellowish to dirty yellow and brown. Oil pollution widely varied from 255 to 16,893 mg of mineral oil per kg of the bog mass surface. According to our studies (2014–2017), the concentration of nitric nitrogen in the soils of the Ershov oil field varied from less than 1 to 3.9 mg kg$^{-1}$, and phosphorus from less than 5 to 190 mg kg$^{-1}$ at individual research points. The surface water of the bog contains low concentrations of nitric nitrogen in range 0.02–2.1 mg L$^{-1}$ (at some points in some years), an average of 1.2 mg L$^{-1}$. The phosphate content in surface waters varies significantly and reaches high values (up to 5.6 mg L$^{-1}$), with an average value of 0.3 mg L$^{-1}$.

In the bogs of the Ershov oil field, 91 species, varieties, and forms of algae (hereinafter referred to as species) were identified from 53 genera, 34 families, 11 classes, and seven taxonomic divisions (Table 2). According to the number of species, charophyte algae prevail with 27%, followed by diatoms with 25%, then cyanobacteria with 18%, and euglenoids with 13%. The lowest richness was found in golden and green algae, which contain 8% each of the total species list (Table 3), and one species of yellow-green algae.

**Table 2.** Species composition of algae and cyanobacteria in the studied sites of the bogs of the Ershov oil field in the Vakh River basin of the Khanty-Mansiysk Autonomous Okrug–Yugra (KMAO-Yugra).

| Taxa | 1 | 2 | 3 | 4 | 5 | 6 | 7 | Hab | T | Oxy | pH | Sal | Wat | Sap | S | Tro | Aut-Het |
|---|---|---|---|---|---|---|---|---|---|---|---|---|---|---|---|---|---|
| **Cyanobacteria** | | | | | | | | | | | | | | | | | |
| *Amorphonostoc paludosum* (Kützing ex Bornet and Flahault) Elenkin | - | 1 | - | - | - | - | - | P-B, S | - | st | - | - | - | b-o | 1.6 | m | - |
| *Anabaena minutissima* Lemmermann | - | - | - | 1 | - | - | - | B | - | - | - | - | - | - | - | - | - |
| *Anabaena* sp. 1 | - | - | - | - | - | - | 1 | - | - | - | - | - | - | - | - | - | - |
| *Anathece clathrata* (West and G.S. West) Komárek, Kastovsky and Jezberová | - | - | - | 1 | - | - | - | P | - | - | - | hl | - | o-a | 1.8 | me | - |
| *Aphanizomenon flosaquae* Ralfs ex Bornet and Flahault | - | - | - | - | - | - | 1 | P | - | - | - | hl | - | o-a | 1.95 | m | - |
| *Aphanizomenon* sp. | - | - | - | - | - | - | 1 | - | - | - | - | - | - | - | - | - | - |
| *Chroococcus turgidus* (Kützing) Nägeli | - | - | - | 1 | - | - | - | P-B, S | - | aer | alf | hl | - | x-b | 0.8 | - | - |
| *Coelosphaerium dubium* Grunow | - | - | 1 | - | - | - | - | - | - | - | - | - | - | - | - | - | - |
| *Coelosphaerium kuetzingianum* Nägeli | - | - | 1 | - | - | - | - | P | - | - | - | i | - | b-o | 1.6 | m | - |
| *Cylindrospermum michailovskoense* Elenkin | - | - | - | - | - | - | 1 | B | - | - | - | - | - | - | - | me | - |
| *Gloeocapsopsis magma* (Brébisson) Komárek and Anagnostidis ex Komárek | - | - | - | 1 | - | - | - | S | - | - | ind | i | - | - | - | - | - |
| *Leptobasis striatula* (F.C. Hy) Elenkin | 1 | - | - | - | - | - | - | - | - | - | - | - | - | - | - | - | - |
| *Merismopedia tenuissima* Lemmermann | 1 | - | - | - | - | - | - | P-B | - | - | - | hl | - | b-a | 2.4 | e | - |
| *Microcystis aeruginosa* (Kützing) Kützing | - | - | 1 | - | - | - | - | P | - | - | - | hl | - | b | 2.1 | e | - |
| *Nodularia spumigena* Mertens ex Bornet and Flahault | - | 1 | - | - | - | - | - | B, S | - | - | - | - | - | o-a | 1.8 | - | - |
| *Scytonema coactile* Montagne ex Bornet and Flahault | - | 1 | - | - | - | - | - | - | - | - | - | - | - | b | 2 | - | - |
| *Synechocystis crassa* Woronichin | - | 1 | - | - | - | - | - | - | - | - | - | - | - | - | - | - | - |
| **Ochophyta (Chrysophyta)** | | | | | | | | | | | | | | | | | |
| *Chrysococcus rufescens* Klebs | 1 | - | - | - | - | - | - | - | - | - | - | hb | - | o-b | 1.4 | - | - |
| *Dinobryon cylindricum* O.E.Imhof | - | - | 1 | - | - | - | - | P | - | - | - | i | - | o | 1.2 | - | - |
| *Dinobryon divergens* O.E.Imhof | - | - | 1 | - | - | 1 | - | P | - | st-str | ind | i | - | o-b | 1.45 | - | - |
| *Dinobryon pediforme* (Lemmermann) Steinecke | - | - | - | - | - | 1 | - | P | - | - | - | - | - | o | 1.2 | - | - |
| *Dinobryon sertularia* Ehrenberg | 1 | 1 | - | - | - | 1 | - | P | - | - | - | i | - | o | 1.3 | - | - |
| *Dinobryon sociale* (Ehrenberg) Ehrenberg | - | - | - | - | - | 1 | - | P | - | - | - | i | - | o | 1.2 | - | - |
| *Mallomonas* sp. | - | - | 1 | - | - | - | - | - | - | - | - | - | - | - | - | - | - |
| **Bacillariophyta** | | | | | | | | | | | | | | | | | |
| *Achnanthhes* sp. | 1 | - | - | - | - | - | - | - | - | - | - | - | - | - | - | - | - |
| *Cyclotella meneghiniana* Kützing | 1 | - | - | - | - | - | - | P-B | temp | st | alf | hl | sp | a-o | 2.8 | e | hne |
| *Epithemia gibba* (Ehrenberg) Kützing | - | 1 | 1 | - | - | - | - | B | temp | - | alb | i | es | x-o | 0.4 | - | - |
| *Eunotia arcus* Ehrenberg | 1 | - | - | - | - | 1 | - | B | - | st-str | acf | i | - | x-o | 0.5 | ot | ats |
| *Eunotia exigua* (Brébisson ex Kützing) Rabenhorst | 1 | - | - | - | 1 | - | - | P-B | - | st-str | acf | hb | es | x-o | 0.45 | o-e | ate |
| *Eunotia lunaris* (Ehrenberg) Grunow var. *lunaris* | 1 | 1 | 1 | - | 1 | 1 | - | B | - | st | ind | i | - | o | 1 | - | - |
| *Eunotia lunaris* var. *capitata* (Grunow) Schönfeldt | - | 1 | - | - | 1 | - | - | B | - | st | ind | i | - | - | - | - | - |
| *Eunotia lunaris* var. *subarcuata* (Nägeli ex Kützing) Grunow | - | 1 | - | - | - | - | - | B | - | st | ind | i | - | o | 1 | - | - |

**Table 2.** *Cont.*

| Taxa | 1 | 2 | 3 | 4 | 5 | 6 | 7 | Hab | T | Oxy | pH | Sal | Wat | Sap | S | Tro | Aut-Het |
|---|---|---|---|---|---|---|---|---|---|---|---|---|---|---|---|---|---|
| *Eunotia major* (W.Smith) Rabenhorst | - | - | 1 | - | - | - | - | B | - | - | acf | hb | - | x-o | 0.4 | - | - |
| *Eunotia monodon* Ehrenberg | - | - | 1 | - | - | - | - | B | - | st-str | acf | hb | - | x-o | 0.4 | ot | ats |
| *Eunotia neocompacta* S.Mayama | 1 | - | - | - | 1 | - | - | B | - | - | - | i | - | x-o | 0.5 | - | - |
| *Eunotia parallela* Ehrenberg | - | 1 | 1 | - | - | - | - | P-B | - | str | acf | i | - | x | 0.3 | ot | ats |
| *Eunotia tenella* (Grunow) Hustedt | - | - | - | - | 1 | - | - | B | - | str | acf | hb | es | o-x | 0.7 | ot | ats |
| *Eunotia* sp. | 1 | - | - | - | - | - | - | - | - | - | - | - | - | x-o | 0.4 | - | - |
| *Frustulia saxonica* Rabenhorst | - | 1 | - | - | - | - | - | B | - | st | acf | hb | es | x | 0.3 | ot | ats |
| *Hantzschia amphioxys* var. *constricta* Pantocsek | 1 | - | - | - | - | - | - | B | - | - | ind | i | - | - | - | - | - |
| *Pinnularia interrupta* W.Smith | - | 1 | - | - | - | - | - | B | - | - | ind | i | - | - | - | - | - |
| *Pinnularia major* (Kützing) Rabenhorst | - | - | 1 | - | - | - | - | B | temp | st-str | ind | i | - | o-x | 0.6 | me | ate |
| *Pinnularia subcapitata* W.Gregory | 1 | - | - | - | 1 | - | - | B | - | st-str | ind | i | sp | o-x | 0.6 | o-m | ate |
| *Pinnularia* sp. | - | - | 1 | - | - | - | - | - | - | - | - | - | - | - | - | - | - |
| *Tabellaria fenestrata* (Lyngbye) Kützing | - | - | 1 | - | - | - | - | P-B | - | st-str | ind | i | es | x | 0.3 | o-m | ats |
| *Tabellaria flocculosa* (Roth) Kützing | - | - | 1 | - | - | - | - | P-B | eterm | st-str | acf | i | es | o-x | 0.6 | ot | ats |
| *Ulnaria ulna* (Nitzsch) Compère | - | 1 | - | - | - | - | - | P-B | temp | st-str | ind | i | es | b | 2.25 | o-e | ate |
| **Euglenophyta** | | | | | | | | | | | | | | | | | |
| *Astasia curvata* (G.A.Klebs) G.A.Klebs | - | - | - | 1 | - | - | - | P-B | eterm | st-str | - | - | - | a | 3.4 | - | - |
| *Astasia sagittifera* Skuja | - | - | - | 1 | - | - | - | P | cool | st-str | - | - | - | a | 3 | - | - |
| *Euglena brevis* P.Christ | - | - | - | 1 | - | - | - | - | - | - | - | - | - | - | - | - | - |
| *Euglena pisciformis* Klebs | - | - | - | 1 | - | 1 | - | P-B | eterm | st-str | alf | mh | - | a | 3 | - | - |
| *Eutreptia lanowii* Steuer | - | - | - | - | - | - | 1 | - | - | - | - | mh | - | - | - | - | - |
| *Lepocinclis steinii* (Lemmermann) Lemmermann | - | - | - | - | - | - | 1 | P | eterm | st | ind | i | - | b | 2.2 | - | - |
| *Menoidium tortuosum* (Stokes) Lemmermann | - | - | - | - | - | - | 1 | P-B | eterm | st-str | ind | - | - | a | 3 | - | - |
| *Phacus oscillans* G.A.Klebs | - | - | - | - | - | - | 1 | P-B | - | st-str | ind | i | - | - | - | - | - |
| *Strombomonas acuminata* var. *verrucosa* Teodoresco | - | - | 1 | - | - | - | - | P | - | st-str | ind | i | - | b | 2.2 | - | - |
| *Trachelomonas oblonga* Lemmermann | 1 | - | - | - | - | - | - | P | eterm | st-str | - | i | - | b-a | 2.4 | - | - |
| *Trachelomonas planctonica* Svirenko | 1 | 1 | - | - | - | - | 1 | P | eterm | st-str | ind | i | - | b | 2.1 | - | - |
| *Trachelomonas volvocina* (Ehrenberg) Ehrenberg | 1 | - | 1 | - | - | 1 | - | B | eterm | st-str | ind | i | - | b | 2 | - | - |
| **Ochrophyta (Xanthophyceae)** | | | | | | | | | | | | | | | | | |
| *Tribonema spirotaenia* Ettl | - | - | 1 | - | - | - | - | - | - | - | - | - | - | - | - | - | - |
| **Chlorophyta** | | | | | | | | | | | | | | | | | |
| *Coelastrum microporum* Nägeli | 1 | - | 1 | - | - | - | - | P-B | - | st-str | ind | i | - | b | 2.3 | - | - |
| *Coenococcus planctonicus* Korshikov | - | - | 1 | - | 1 | - | - | P | - | - | - | i | - | o-b | 1.5 | - | - |
| *Kirchneriella obesa* (West) West and G.S. West | - | - | - | 1 | - | - | - | - | - | - | - | - | - | - | - | - | - |
| *Mucidosphaerium pulchellum* (H.C.Wood) C.Bock, Proschold and Krienitz | 1 | - | - | 1 | - | - | - | P-B | - | st-str | ind | i | - | b | 2.3 | - | - |
| *Oocystis rhomboidea* Fott | 1 | - | - | - | - | - | - | P | - | - | - | - | - | o-a | 1.8 | - | - |
| *Sphaerocystis planctonica* (Korshikov) Bourrelly | - | - | 1 | - | - | - | - | - | - | - | - | - | - | o | 1 | - | - |
| *Tetradesmus obliquus* (Turpin) M.J.Wynne | 1 | - | - | 1 | - | - | 1 | P-B | - | st-str | ind | i | - | b | 2.05 | - | - |

**Table 2.** *Cont.*

| Taxa | 1 | 2 | 3 | 4 | 5 | 6 | 7 | Hab | T | Oxy | pH | Sal | Wat | Sap | S | Tro | Aut-Het |
|---|---|---|---|---|---|---|---|---|---|---|---|---|---|---|---|---|---|
| **Charophyta** | | | | | | | | | | | | | | | | | |
| *Actinotaenium cucurbitinum* (Bisset) Teiling | 1 | - | - | - | - | - | - | P-B | - | - | acf | - | - | - | - | o-m | - |
| *Actinotaenium globosum* (Bulnheim) Kurt Förster ex Compère | - | - | - | 1 | - | - | - | P-B | - | - | acf | - | - | - | - | o-m | - |
| *Actinotaenium rufescens* (Cleve) Teiling | 1 | 1 | 1 | 1 | - | - | 1 | B | - | - | acf | - | - | - | - | m | - |
| *Actinotaenium spinospermum* (Joshua) Kouwets and Coesel | - | - | - | - | - | 1 | - | B | - | - | acf | - | - | - | - | m | - |
| *Actinotaenium wollei* (West & G.S.West) Teiling ex Ruzika and Pouzar | - | - | - | - | 1 | - | - | B | - | - | acf | - | - | - | - | o | - |
| *Bambusina borreri* (Ralfs) Cleve | - | - | 1 | - | 1 | 1 | - | P-B | - | - | acf | - | - | x-b | 0.9 | o | - |
| *Closterium acutum* Brébisson | - | - | - | - | - | 1 | - | P-B | - | st-str | ind | - | - | b | 2.05 | m | - |
| *Closterium baillyanum* (Brébisson ex Ralfs) Brébisson | 1 | - | - | - | - | - | - | B | - | - | ind | - | - | - | - | o-m | - |
| *Closterium calosporum* var. *brasiliense* Børgesen | 1 | - | - | - | - | - | - | B | - | - | acf | - | - | - | - | o-m | - |
| *Closterium closterioides* (Ralfs) A.Louis and Peeters var. *closterioides* | - | - | 1 | - | - | - | - | B | - | - | acf | - | - | o-x | 0.6 | o-m | - |
| *Closterium closterioides* var. *intermedium* (J.Roy and Bisset) Ruzicka | 1 | - | - | - | - | - | - | B | - | - | acf | - | - | o-x | 0.7 | o-m | - |
| *Closterium dianae* Ehrenberg ex Ralfs | 1 | - | - | - | - | - | - | P-B | - | st-str | acf | - | - | x-b | 0.8 | m | - |
| *Closterium striolatum* Ehrenberg ex Ralfs | - | 1 | - | - | - | - | - | P-B | - | - | acf | - | - | o | 1.2 | o-m | - |
| *Desmidium swartzii* C.Agardh ex Ralfs | - | 1 | - | - | - | - | - | B | - | - | ind | i | - | o-x | 0.6 | m | - |
| *Euastrum ansatum* Ehrenberg ex Ralfs | - | 1 | - | - | - | - | - | P-B | - | - | acf | - | - | x-o | 0.5 | o-m | - |
| *Euastrum divergens* var. *ornatum* (O.Borge) Schmidle | 1 | - | - | - | - | - | - | - | - | - | - | - | - | - | - | - | - |
| *Euastrum dubium* var. *pseudocambrense* Grönblad | 1 | - | - | - | - | - | - | - | - | - | - | - | - | - | - | - | - |
| *Euastrum lapponicum* Schmidle | - | - | 1 | - | - | - | - | P | - | - | ind | hb | - | x-o | 0.5 | m | - |
| *Euastrum oblongum* Ralfs | - | - | 1 | - | - | - | - | B | - | - | acf | - | - | o-x | 0.6 | m | - |
| *Haplotaenium minutum* (Ralfs) Bando | - | - | - | - | - | 1 | - | B | - | - | acf | - | - | x-o | 0.5 | m | - |
| *Octacanthium bifidum* (Brébisson) Compère | - | - | - | - | - | 1 | - | B | - | - | acf | - | - | - | - | o-m | - |
| *Spirogyra* sp. | - | 1 | - | - | - | - | - | B | - | - | - | - | - | - | - | - | - |
| *Staurastrum dilatatum* Ehrenberg ex Ralfs | 1 | 1 | 1 | - | 1 | - | - | P | - | - | - | - | - | - | - | - | - |
| *Staurastrum gracile* Ralfs ex Ralfs | - | - | 1 | - | - | - | - | P-B | - | st | acf | i | - | o | 1.3 | m | - |
| *Staurastrum ralfsii* var. *depressum* (J. Roy and Bisset) Coesel and Meesters | 1 | - | - | - | - | - | - | B | - | - | ind | - | - | o | 1.3 | m | - |

Note. Abbreviation of the ecological groups: Substrate—S, soil; B, benthic as a whole; P-B, planktonic-benthic; P, planktonic. Temperature—cool, cool loving; temp, temperate temperature waters; eterm, eurythermal; warm, warm waters. Water disturbance and oxygenation—aer, aerophiles; str, streaming well oxygenated waters; st-str, low streaming, middle oxygenated waters; st, standing, low oxygenated waters. Water pH—acf, acidophiles; ind, pH-indifferents; alf, alkaliphiles; alb, alkalibiontes. Salinity—hb, halophobes; i, chloride-tolerated (indifferent); hl, halophiles; mh, mesohalobes. Saprobity according to Watanabe—es, eurysaprobes; sp, saprophiles. Class of water quality—Class 1-4 according to EU FWD. Trophy—ot, oligotraphentes; o-m, oligo-mesotraphhentes; m, mesotraphentes; me, meso-eutraphentes; e, eutraphentes; o-e, from oligo- to eutraphentes. Nutrition type—ats, strongly autotropes; ate, autotrophic withstand low nitrogen load; hne, particular heterotrophes (mixotrophes).

**Table 3.** Calculation results of taxonomic and bioindication analysis of algae and cyanobacteria composition in the studied sites of the bogs of the Ershov oil field in the Vakh River of the KMAO-Yugra.

| Variable | 1 | 2 | 3 | 4 | 5 | 6 | 7 |
|---|---|---|---|---|---|---|---|
| **Division** | | | | | | | |
| Bacillariophyta | 9 | 8 | 9 | 0 | 6 | 2 | 0 |
| Charophyta | 10 | 6 | 7 | 2 | 3 | 5 | 1 |
| Chlorophyta | 4 | 0 | 3 | 3 | 1 | 0 | 1 |
| Cyanobacteria | 2 | 3 | 1 | 0 | 0 | 0 | 0 |
| Euglenophyta | 3 | 1 | 2 | 4 | 0 | 2 | 5 |
| Ochophyta (Chrysophyta) | 3 | 1 | 1 | 0 | 0 | 1 | 1 |
| Ochrophyta (Xanthophyceae) | 0 | 0 | 1 | 0 | 0 | 0 | 0 |
| **Abundance, $10^3$ cells L$^{-1}$** | **40** | **7** | **36** | **1** | **3** | **5** | **1** |
| **No. of Species** | **31** | **19** | **24** | **9** | **10** | **10** | **8** |
| **Substrate** | | | | | | | |
| S | 0 | 0 | 0 | 1 | 0 | 0 | 0 |
| B | 11 | 10 | 9 | 2 | 6 | 6 | 2 |
| P-B | 8 | 5 | 6 | 6 | 2 | 3 | 3 |
| P | 5 | 3 | 8 | 2 | 2 | 4 | 3 |
| **Temperature** | | | | | | | |
| cool | 0 | 0 | 0 | 1 | 0 | 0 | 0 |
| temp | 1 | 2 | 2 | 0 | 0 | 0 | 0 |
| eterm | 3 | 1 | 2 | 2 | 0 | 2 | 3 |
| **Oxygenation** | | | | | | | |
| st | 2 | 5 | 2 | 0 | 2 | 1 | 1 |
| st-str | 10 | 2 | 8 | 5 | 2 | 5 | 4 |
| str | 0 | 1 | 1 | 0 | 1 | 0 | 0 |
| aer | 0 | 0 | 0 | 1 | 0 | 0 | 0 |
| **pH** | | | | | | | |
| acf | 7 | 5 | 9 | 2 | 4 | 5 | 1 |
| ind | 10 | 7 | 8 | 3 | 3 | 4 | 5 |
| alf | 1 | 0 | 0 | 2 | 0 | 1 | 0 |
| alb | 0 | 1 | 1 | 0 | 0 | 0 | 0 |
| **Salinity** | | | | | | | |
| hb | 2 | 1 | 3 | 0 | 2 | 0 | 0 |
| i | 12 | 10 | 14 | 3 | 5 | 6 | 4 |
| hl | 2 | 0 | 1 | 2 | 0 | 0 | 1 |
| mh | 0 | 0 | 0 | 1 | 0 | 1 | 1 |
| **Watanabe** | | | | | | | |
| es | 1 | 3 | 3 | 0 | 2 | 0 | 0 |
| sp | 2 | 0 | 0 | 0 | 1 | 0 | 0 |
| **Class of Water quality** | | | | | | | |
| Class 1 | 4 | 4 | 6 | 0 | 2 | 2 | 0 |
| Class 2 | 7 | 5 | 11 | 1 | 5 | 6 | 0 |
| Class 3 | 8 | 5 | 5 | 3 | 0 | 2 | 4 |
| Class 4 | 1 | 0 | 0 | 3 | 0 | 1 | 0 |
| **Trophic state** | | | | | | | |
| ot | 1 | 2 | 4 | 0 | 1 | 1 | 0 |
| o-m | 5 | 2 | 2 | 1 | 1 | 1 | 0 |
| m | 3 | 3 | 5 | 1 | 0 | 3 | 2 |
| me | 0 | 0 | 1 | 1 | 0 | 0 | 1 |
| e | 2 | 0 | 1 | 0 | 0 | 0 | 0 |
| o-e | 1 | 1 | 0 | 0 | 1 | 0 | 0 |
| **Nutrition type** | | | | | | | |
| ats | 1 | 2 | 4 | 0 | 1 | 1 | 0 |
| ate | 2 | 1 | 1 | 0 | 2 | 0 | 0 |
| hne | 1 | 0 | 0 | 0 | 0 | 0 | 0 |
| **Index saprobity S** | **1.50** | **1.07** | **0.91** | **2.03** | **0.73** | **1.33** | **2.13** |

The ecological group's abbreviation as in Table 2. The Classes of water quality colored by EU color scale.

The most diverse was the community in sites 1 and 3 with 31 and 24 species, respectively (Tables 2 and 3), where the Charophyta species prevail.

Site 1 was selected as a control located away from the source of oil pollution. Direct exposure to oil and oil products is not recorded here. The water temperature during the study period reached 21 °C; pH was 4.1–4.8, and the oil content of 255.65 ± 1.50 mg kg$^{-1}$. The composition of the algal community was green-diatom. Two species of algae from the Charophyta division dominate the community: *Closterium baillyanum* (30.3 × 10$^3$ cells L$^{-1}$) and *Actinotaenium cucurbitinum* (1.8 × 10$^3$ cells L$^{-1}$).

On site 2, the water temperature was 21 °C, pH 5.4, and the low content of petroleum products (419.40 ± 1.30 mg kg$^{-1}$) has been revealed. The samples were enriched by the large-cell algae of the diatom group represented by the genera *Pinnularia* and *Eunotia*, which are usually part of the bog complex. Two species dominate: with an abundance of 3.0 × 10$^3$ cells L$^{-1}$ *Rhopalodia gibba* from diatoms and *Spirogyra* sp. (1.5 × 10$^3$ cells L$^{-1}$) from Charophyta. The last one species was not determined due to lack of conjugated filaments.

On site 3, the water temperature was 22 °C, pH 4.0–4.5, and defined the medium content of oil products (1203.10 ± 25.00 mg kg$^{-1}$). The composition of the algal community was green-diatom. As part of the dominants were cyanobacteria *Microcystis aeruginosa*, which abundance was of 15 × 10$^3$ cells L$^{-1}$ and charophyte alga *Bambusina borreri* with 9.9 × 10$^3$ cells L$^{-1}$.

The water temperature 22 °C, pH 4.6 and the medium concentration of petroleum products (1500.38 ± 21.20 mg kg$^{-1}$) has been revealed on site 4. The algae community was of low species-rich, small-celled, with a low abundance. In the microscope field of view, spores of cyanobacteria and small coccoid algae with a diameter of 2–7 microns were observed. The algae complex was represented by cyanobacteria, euglenoid, and green algae. The total number of algae, excluding the above-described small-celled coccoid algae, was 24.1 × 103 cells L$^{-1}$. Dominant was the cyanobacteria *Anabaena minutissima* with abundance of 18.75 × 10$^3$ cells L$^{-1}$, which is 76.8% of the total abundance in the community of site 4.

Three taxonomic divisions represented the community of site 5. Water temperature was 18 °C, pH 4.3, and the average value of oil concentration defined (1715.47 ± 4.56 mg kg$^{-1}$). The algal complex was diatom-green. Dominants are two species of diatoms indicators of an acidic environment: *Eunotia lunaris* var. *capitata* (1.1 × 10$^3$ cells L$^{-1}$) and *Pinnularia subcapitata* var. *hilseana* (0.4 × 10$^3$ cells L$^{-1}$).

At the studied site 6, the water temperature was 22 °C, pH 4.2, and found the high concentration of oil products (3308.5 ± 5.33 mg kg$^{-1}$). Four divisions were represented in the algae community. *Dinobryon divergens* from Chrysophyta and *Octacanthium bifidum* from Charophyta dominate (3.0 and 2.5 × 10$^3$ cells L$^{-1}$, respectively).

At the most contaminated site 7, the water temperature was 21 °C, the pH 4.1, and the highest oil concentration (16,893.80 ± 2.54 mg kg$^{-1}$). Flatworms are defined in the samples. The composition of algae includes four divisions. The algal complex is euglenoid-green, with small forms of algae observed, as well as spores of cyanobacteria. The highest abundance have euglenoid algae *Eutreptia lanowii* (0.5 × 10$^3$ cells L$^{-1}$), *Phacus oscillans* (0.33 × 10$^3$ cells L$^{-1}$), and *Lepocinclis steinii* (0.23 × 10$^3$ cells L$^{-1}$).

Bioindication results are represented in Table 3. There can be seen a prevalence of benthic eurythermic forms, which preferred standing acidic waters with low salinity and are middle saturated by nutrients. The algal communities were represented by oligo-mesotrophic autotrophic species that can survive in the waters of Class 2 water quality. Index saprobity S varied from 0.73 on the site 5 to 2.13 on most contaminated site 7.

The difference between distributions of the bioindicator's data and the chemical data in the studied sites can be better recognized with help of the ecological maps that were constructed in the Statistica 12.0 program on the base of Tables 1 and 3. Figures 3–7 show the distribution of chemical variables and of the algae species-indicators across the studied sites. It can be seen that water pH was higher in the eastern part of the bogs of the Ershov oil field (Figure 3a), the oil concentration has high values in the station 7 with decreasing of the value in the periphery (Figure 3b). Species richness

of algae community and abundance were maximal on control site 1 on the northeastern part of bog (Figure 3c,d).

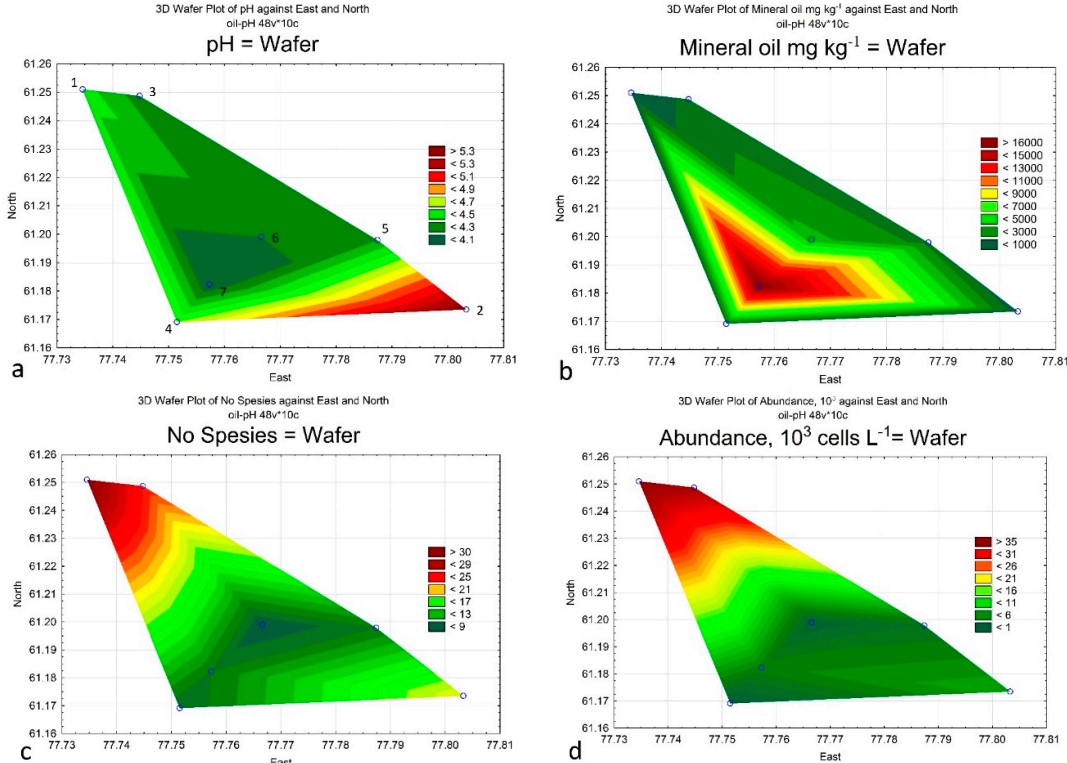

**Figure 3.** Distribution maps for water pH (**a**), mineral oil (**b**), species richness (**c**), and algal abundance (**d**) in seven studied sites of the bogs of the Ershov oil field in the Vakh River basin of the KMAO-Yugra.

Distribution of the species richness in most represented divisions show differences in preferences of environmental properties of studied sites. As shown, diatoms in communities avoided the contaminated area (Figure 4a), green algae were richest in control site (Figure 4b), cyanobacteria tolerated high pH levels well (Figure 4c), and only euglenoids have been distributed near the contaminated area (Figure 4d).

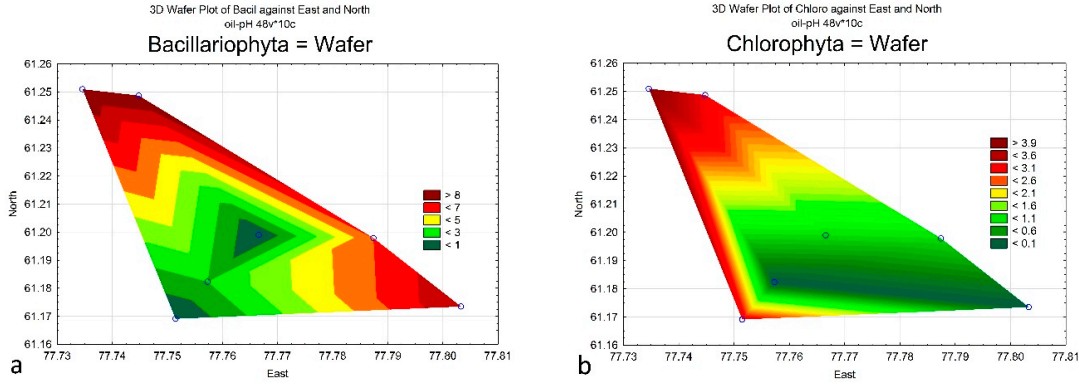

**Figure 4.** *Cont*.

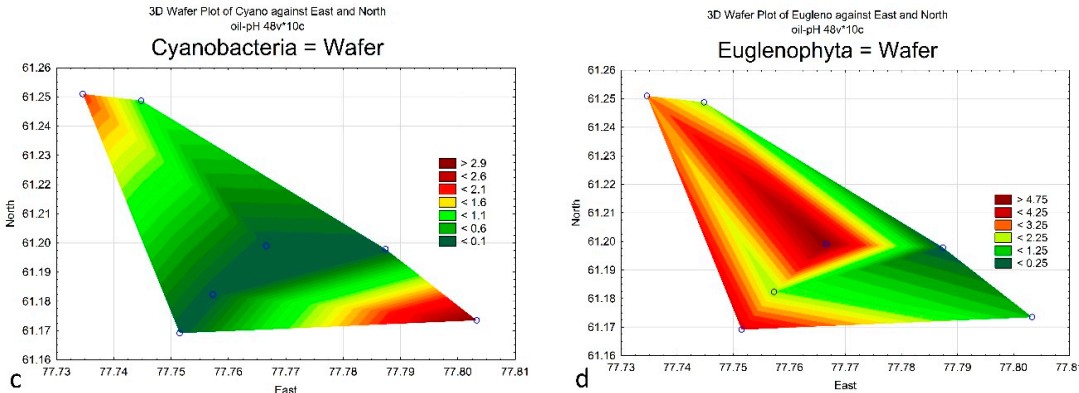

**Figure 4.** Distribution maps for species richness of Bacillariophyta (**a**), Chlorophyta (**b**), Cyanobacteria (**c**), and Euglenophyta (**d**) algae in seven studied sites of the bogs of the Ershov oil field in the Vakh River basin of the KMAO-Yugra.

Indicators of water temperature marked the area in the southwestern part of the bogs of the Ershov oil field as the coolest one (Figure 5a), and all western areas as temperate in temperature (Figure 5b). Water salinity is a very important variable for the bogs and the maps show that halophobic species preferred control site (Figure 5c), whereas mesohalobes in contrary were richest near the contaminated area (Figure 5d).

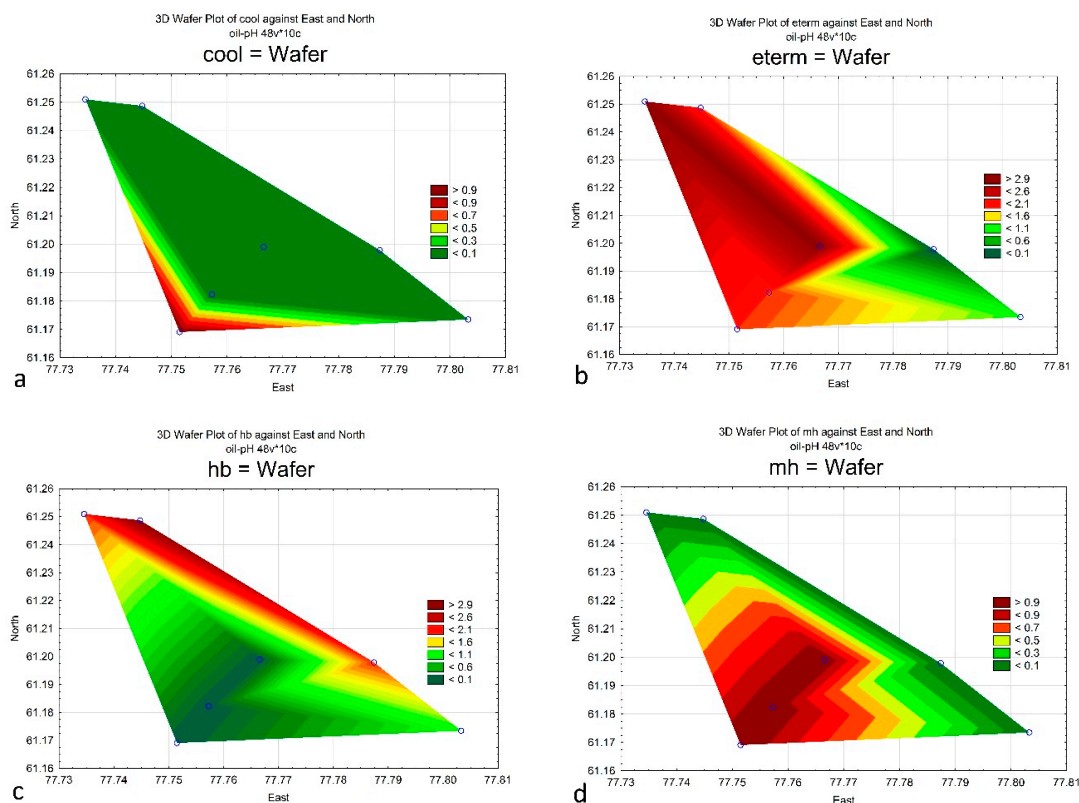

**Figure 5.** Distribution maps for species indicators of cool water (**a**), eurythermic water (**b**), halophobes (**c**), and mesohalobes (**d**) in seven studied sites of the bogs of the Ershov oil field in the Vakh River basin of the KMAO-Yugra.

Water pH in the bogs is usually low than 7, therefore, the distribution of indicators of high and low acidity waters can help to recognize the acidification areas. Figure 6a,b demonstrated that acidophilic species were richest in the control site, but species that preferred water with pH more than 7 are concentrated in communities of the southwestern part of the bogs of the Ershov oil field. The

ecological niche of the alkaliphilic species is usually wide and therefore they small number can be included as a part of the acidophilic community. In any case, the presence of high-level pH indicators in the bog community can indicate the anthropogenic disturbance of community habitat. The nutrition type indication demonstrates that truly autotrophic species avoid the contaminated area (Figure 6c). The trophic state indicators characterize the studied environment as consisting of oligotrophic and oligo-eutrophic waters that avoid the contaminated area (Figure 6d,e). All these anomalies in the distribution of indicator species indicate a disturbance in the habitat and a deviation of its indicators from those typical for the nature of the region. The most informative variable of organic pollution is Index saprobity S. The map of Index S distribution (Figure 6f) shows the highest value near the contaminated site and that it is very similar to the Euglenophyta species distribution (Figure 4d).

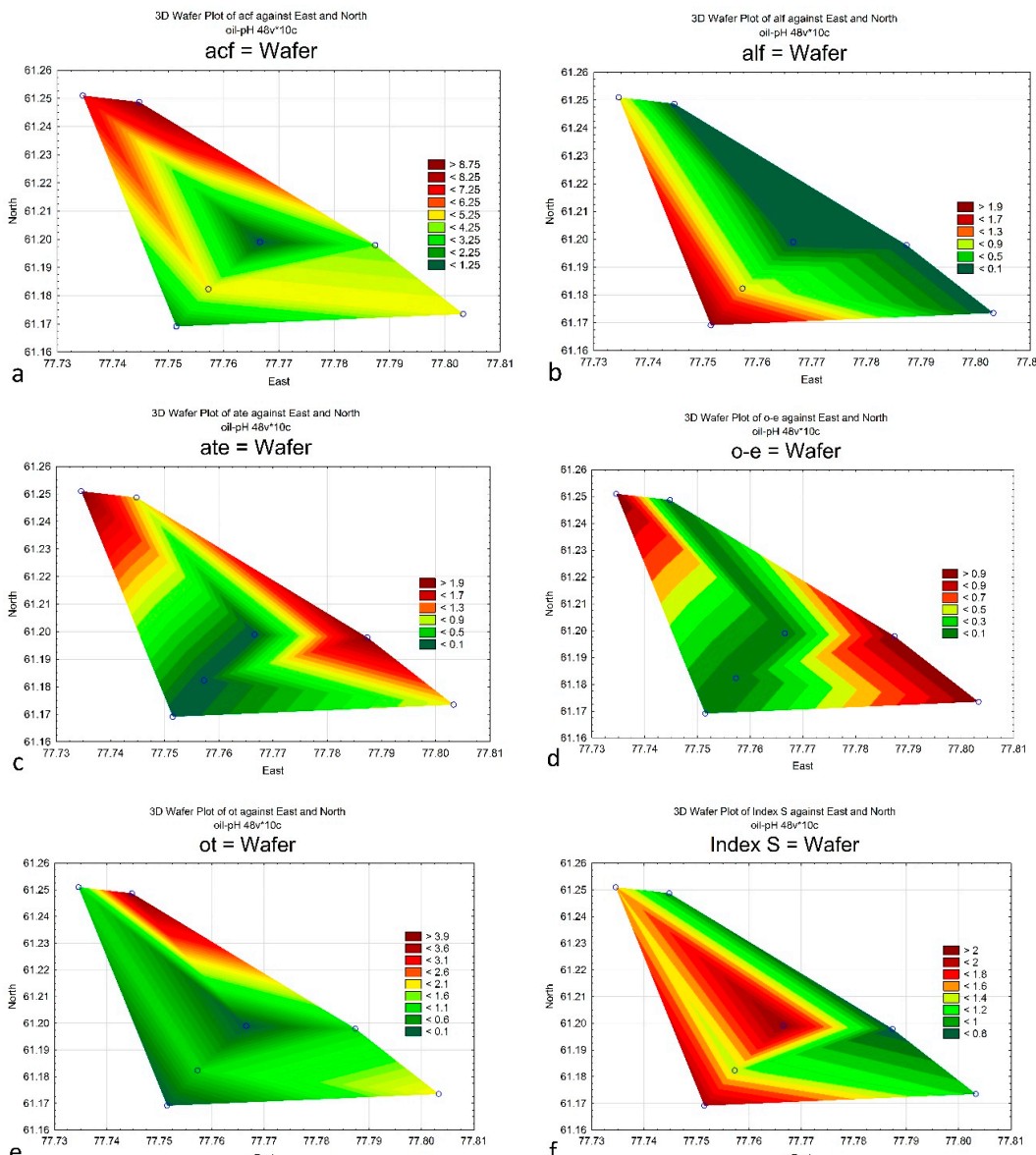

**Figure 6.** Distribution maps for Index saprobity S and species indicators: acidophiles (**a**), alkaliphiles (**b**), ologo-eutraphentes (**c**) species with autotropic nutrition (**d**), oligotrophic water indicators (**e**), and Index saprobity S (**f**) in seven studied sites of the bogs of the Ershov oil field in the Vakh River basin of the KMAO-Yugra.

The water quality maps are represented in Figure 7. It can be seen that the clean water species all avoid the contaminated area (Figure 7a–c) and preferred the environment in the control site, whereas

indicators of organically polluted waters Class 4 were mostly represented in the southwestern part of the bogs of the Ershov oil field (Figure 7c); nevertheless, the Index saprobity S was maximal in site 7.

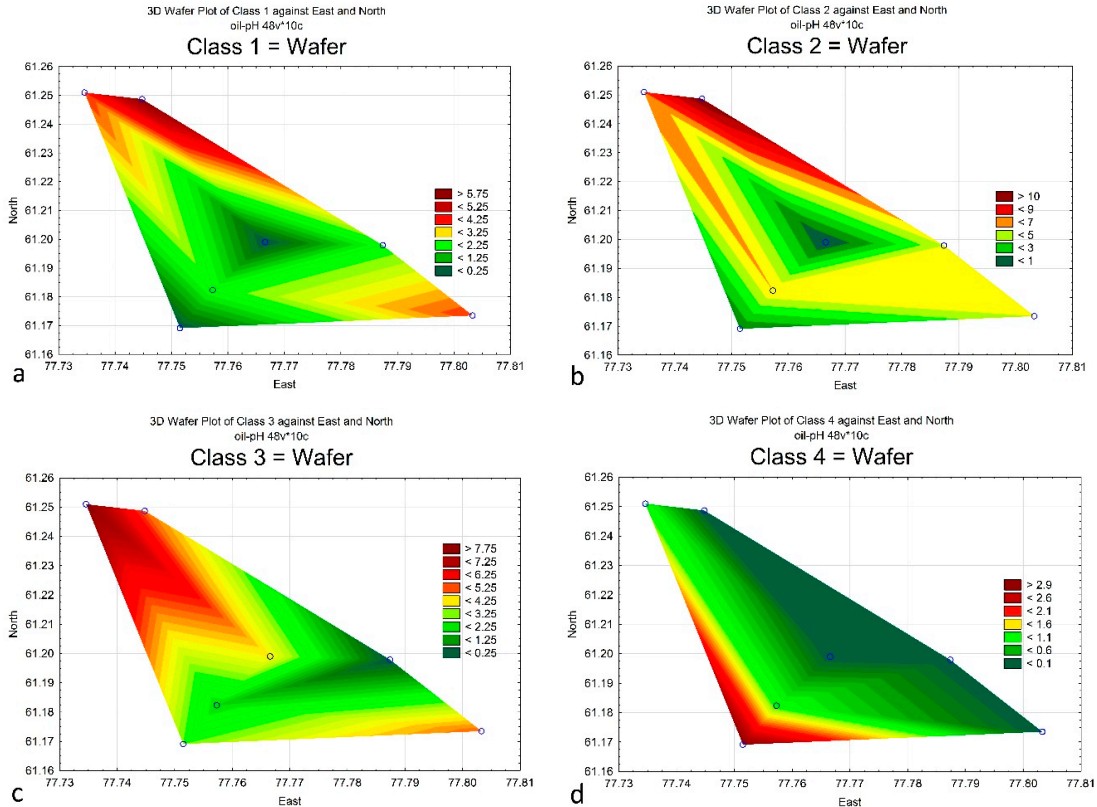

**Figure 7.** Distribution maps for indicators of Water Quality Class 1–4 (**a**–**d**) in seven studied sites of the bogs of the Ershov oil field in the Vakh River basin of the KMAO-Yugra.

We tried to reveal the similarity of the site communities of the bogs of the Ershov oil field in the statistic program R. Figure 8 shows that the most similar were communities of the sites 2, 3, and 5 (blue lines), whereas all other communities were with negative correlation (rose lines). It is noteworthy that sections 2, 3, and 5 are located on the periphery of the contaminated area (Figure 1).

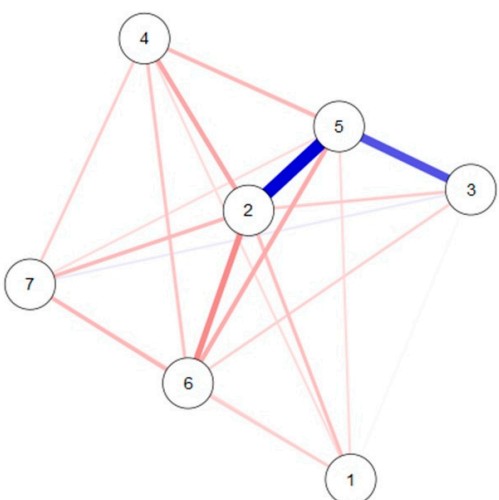

**Figure 8.** JASP (Jeffreys's Amazing Statistics Program) plot of the algal communities similarities in seven studied sites of the bogs of the Ershov oil field in the Vakh River basin of the KMAO-Yugra. Number of sites as in Table 1.

Our calculation of similarity in the BioDiversity Pro program (Table 4) confirms these results and shows the highest value of the Pearson coefficient for the communities of sites 2 and 5 situated in the southeast from the contaminated site 7. Table 4 also shows low similarity values for all studied communities, and so we decide here to reveal which communities are most individual. Table 5 represented the percentage of unique species in the algae and cyanobacteria communities of the Ershov oil field. It can be seen that the most specific was the control site community with 68% of unique species, in contrary to the community of contaminated site 7 without unique species.

**Table 4.** Similarity matrix for the species richness in studied sites of the bogs of the Ershov oil field in the Vakh River basin of the KMAO-Yugra. Highest coefficient of similarity is marked by bold.

| Site | 1 | 2 | 3 | 4 | 5 | 6 | 7 |
|------|---|------|------|------|-------|------|------|
| 1 |  | 0.97 | 2.11 | 1.12 | 4.87 | 0.84 | 1.79 |
| 2 |  |  | 2.04 | 1.58 | **17.94** | 2.67 | 2.50 |
| 3 |  |  |  | 0.43 | 2.26 | 0.45 | 2.16 |
| 4 |  |  |  |  | 0.00 | 6.06 | 2.08 |
| 5 |  |  |  |  |  | 3.81 | 7.26 |
| 6 |  |  |  |  |  |  | 0 |
| 7 |  |  |  |  |  |  |  |

**Table 5.** Unique species estimator in the algae and cyanobacteria communities in studied sites of the bogs of the Ershov oil field in the Vakh River basin of the KMAO-Yugra.

| Percent of Unique Species | |
|---|---|
| **No of Site** | **Estimator** |
| 1 | 68.0 |
| 2 | 54.6 |
| 3 | 40.6 |
| 4 | 33.2 |
| 5 | 21.2 |
| 6 | 8.2 |
| 7 | 0 |

Relationships between the biological (number of species in the taxonomic Divisions from Table 3) and chemical variables (from Table 1) of studied sites in the Ershov oil field were calculated in the CANOCO program. Figure 9 shows that the environmental variables divided into two different groups. As shown, mineral oil and organic pollution (as Index saprobity S) affected the algal community mostly in site 7, with the prevalence of the euglenoid species as a result (red dashed line). Water pH was only one variable that affected site 2 communities with cyanobacteria flourishing (blue dashed line). Such biological variables as species richness and the abundance of the algal cells that can be taken into account as variables, a high level of which characterizes environments most comfortable for algae development, and which were represented by the third group in Figure 9 (black dashed line). Sites 2 and 5 communities were most abundant and species-rich with a prevalence of diatom and Xanthophyceae algae. Therefore, these sites can be assessed as the most rehabilitated and close to the control site 1.

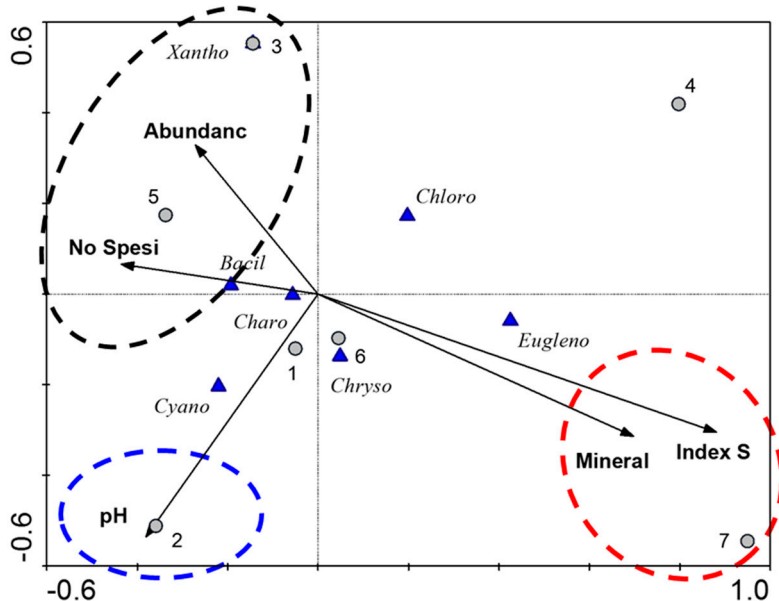

**Figure 9.** Canonical correspondence analysis (CCA) plot for relationships between biological and environmental variables in studied sites of the bogs of the Ershov oil field in the Vakh River basin of the KMAO-Yugra.

## 5. Discussion

It is known [23,24] that autotrophs in the swamps and wetlands have an important role in the process of denitrification. The surface water of the Ershov bog contains insignificant concentrations of nitrates from 0.02 to 2.1 mg L$^{-1}$, an average of 1.2 mg L$^{-1}$ that is in the range of nitrates in the water of the oligotrophic swamps, such as the Everglades (0.1 mg L$^{-1}$) [25] and in Polish swamps (3.0 mg L$^{-1}$) [26].

The Ershov oil deposit bogs' algae species list included 91 taxa belonging to the seven richest taxonomic divisions: Charophyta, Bacillariophyta, Cyanobacteria, and Euglenophyta. Early studies of algal species richness in nearby unpolluted swamps of the region of KMAO-Yugra varied between 76 and 109 taxa [8,11–14]. Therefore, the algae list in the studied Ershov bogs (91) is in the regional range and thus can be used for the analysis. The prevailing charophytic algae in the entire list of studied communities and at the control site especially reflects the natural properties of the species composition of algae of the bog (swamp) complex of species. It is confirmed by the pH of the water in the Ershov oil field, which is acidic, not exceeding 5.4, and these species prefer to inhabit an acidic environment. We cannot significantly compare the divisional distribution in the disturbance gradient in nearby bogs because only a total list of algae had been represented in the references [8,11–15]. Our conclusions for the aquatic communities confirmed a similar tendency with the experimental research where oil pollution caused a rearrangement in the algal community [22], with a change in species composition, and a decrease in the abundance and biomass of soil algae. This experiment revealed that diatoms and yellow-green algae were the most sensitive to oil pollution, and cyanobacteria and green algae were less sensitive, which we confirmed by our research of the natural communities.

In the zone of sphagnum bogs in surrounding areas where direct pollution was not observed, species composition of the identified algae in the bogs of the Agan and Novo-Pokursky deposits, respectively, was 76 and 109 taxa (according to the samples of September 2018) [27]. So, the bogs of the Agan deposits community were dominated by species of diatoms, euglenoids, and green algae: *Coenococcus planctonicus* (724.2 × 10$^3$ cells L$^{-1}$), *Trachelomonas planctoica* (366.6 × 10$^3$ cells L$^{-1}$), and *Rhopalodia gibba* (228.2 × 10$^3$ cells L$^{-1}$) prevailed. In the Novo-Pokursky deposits, the algae community were dominated by diatoms and cyanobacteria species: *Rhopalodia gibba* (765.5 × 10$^3$ cells L$^{-1}$), *Aphanothece clathrata* (630.2 × 10$^3$ cells L$^{-1}$), and *Microcystis grevillei* (352.8 × 10$^3$ cells L$^{-1}$) [27]. In the

Ershov bogs referenced site, Charophyta algae dominated: *Closterium baillyanum* ($30.3 \times 10^3$ cells L$^{-1}$) and *Actinotaenium cucurbitinum* ($1.8 \times 10^3$ cells L$^{-1}$). The numbers are incomparably lower than the abundance of dominants in the nearby bogs. In polluted sites, the algal abundance decreased but the taxonomic spectrum of dominants increased with the addition of diatoms, cyanobacteria, and *Chrysophyta* species.

The important process in the wetlands and swamps is eutrophication, which is defined as the increased availability of elements that limit primary production and is usually associated with surface water pollution in swamps [28]. The organic pollution in the Ershov bogs is expressed in the Indices of saprobity S, which varied in the range 0.73–2.13 in water of Class 2–3 of Water Quality [10]. Therefore, we can conclude, based on nutrient concentration and Index S that the Ershov bogs are still in the oligotrophic state even after the oil pollution impact. At the same time, we revealed increasing in the saline water indicators in the oil-contaminated sites. The results allow us to conclude that the oil production process affected the environment not only with oil but also with saline waters, as was found in the nearby areas [15].

The bioindicator analysis in combination with statistically generated maps that was previously used for the waterbody surface area and for the catchment basin analysis [20] here was used for the first time for the bog surface. A comparison in the distribution of ecological groups and environmental variables with maps helped us to reveal the relationships between oil concentration in the bog surface mass and the distribution of salinity and organic pollution algae indicators. Currently, we do not have comparable data of bioindicator distribution from the nearby bogs because this approach was used for the first time, but the correlation of salinity and nutrients impact is known for the bogs and swamps from different regions [28].

## 6. Conclusions

The gradient of oil pollution in the Ershov oil field of KMAO-Yugra varied between 255 and 16,893 mg of mineral oil per kg of the bog mass surface. The ecosystem sensitivity is based on the acidic water with a pH of about 4.0–5.4. The ecosystems of polluted areas have been rehabilitated but it is difficult to assess their effectiveness. In this way, we have tried to make use of algal bioindicator methods. For the first time, the studied Ershov oil field bogs algae species' list included 91 species, varieties, and forms of algae and cyanobacteria which belonged to seven taxonomic divisions. It has been found that Charophyta algae prevail in studied communities, followed by diatoms, cyanobacteria, and euglenoids. The control site demonstrated significant differences between qualitative and quantitative variables of polluted and unpolluted areas. As revealed with the bioindication and statistical mapping methods, the species richness and abundance of algae were maximal at the control site with the prevalence of green algae, and diatoms algae species strictly avoided occupying oil-contaminated habitats. At the same time, the cyanobacteria were tolerated at a pH close to neutral. The euglenoid algae can survive under oil pollution. Statistical mapping made it possible to point to the southeastern edge of the area of the bogs flooded by cold waters. It is known that the underground waters together with the oil come to the surface, so we indicated it. In our case, the northern bogs, like our studied bogs, have a very sensitive ecosystem. Bioindication showed that halophobic species prefer the specified control site, while salt tolerant species, on the contrary, prefer the contaminated area. Pertaining to organic pollution, we found a similar distribution of species tolerant to organic pollution and Index S was highest in the contaminated site, but clean water indicator species were not found in the contaminated area. Comparative floristic analysis shows the similarity of communities in sites surrounding the contaminated area and in ecosystems under rehabilitation for 10–20 years. The percent of unique species was maximal in the control site, whereas in the oil-polluted areas it was zero. All results in the distribution of indicator species revealed a disturbance in the oil-polluted habitats and a deviation of its indicators from those typical for nature of the region.

Therefore, we can conclude that bioindicator methods not only expanded the possibility of assessing the impact of pollution on the ecosystem of the bogs, but it also revealed environmental

properties that could not be established by other methods, such as in the effectiveness of rehabilitation. Ecological maps of the environmental factors and bioindication results were used for the first time in assessing oil-polluted bogs ecosystem, and therefore can be recommended as methods for the establishing of scientific results in monitoring pollution in the future.

**Author Contributions:** Conceptualization, S.B., O.S.; data curation, O.S., E.Y., and T.S.; formal analysis, S.B., O.S., E.Y., and T.S.; investigation, O.S., E.Y., and T.S.; methodology, S.B. and O.S.; project administration, S.B. and O.S.; resources, O.S., E.Y., and T.S.; supervision, S.B. and O.S.; validation, S.B. and O.S.; visualization, S.B. and E.Y.; writing–original draft, S.B., O.S., E.Y., and T.S.; writing–review and editing, S.B., O.S., E.Y. and T.S.

**Funding:** This research received no external funding.

**Acknowledgments:** The study was carried out with the financial support of the Russian Federal Property Fund and the Government of the Khanty-Mansiysk Autonomous Okrug–Yugra as part of the scientific project No. 18-44-860005. This work is partly supported by the Israel Ministry of Aliyah and Integration.

**Conflicts of Interest:** The authors declare no conflict of interest.

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
