# Peer review of "Bioindication of the Influence of Oil Production on Sphagnum Bogs in the Khanty-Mansiysk Autonomous Okrug–Yugra, Russia"

_diversity, doi:10.3390/d11110207_

Round 1

Reviewer 1 Report

Review of the manuscript “Bioindication of the influence of oil production on sphagnum bogs in the Khanty-Mansiysk Autonomous Okrug – Yugra, Russia” by Olga Skorobogatova, Elvira Yumagulova, Tatiana Storchak.

General comments

The manuscript investigates the species richness and abundance of algae in seven swamps contaminated with oil products in the Khanty-Mansiysk Autonomous Okrug – Yugra, Russia. The study aimed to assess the oil pollution impact on this ecosystem by methods of algal bioindication.

My main concern regards that nutrients are not sufficiently considered. Species richness and abundance of algae are controlled by several environmental factors, and nutrients are among the most important ones. In this study there is no information regarding nutrients, neither from the literature or from the study itself (as no nutrient analyses was performed).

In my opninion, the manuscript is generally nicely written and clear, but before the pubblication the nutrient availability of the study area should be clarifyied, maybe just showing literature data about it (if available). Some further changes are given below.

Specific comments

L77. Here only the site 1 is referred as control (background), but in figure 1 there are showed several background areas. What are those background areas? Other controls sites for those close to them? Please clarify it.

L157. Please change “thousand cells per 1 liter of water” to “x 103 cell L-1”.

L158, 162, 163, 167, 168, 174, 175, 178, 179, 182 and 188. Change “thousand” to “x 103”.

Figures

Figure 1. This figure is not easy to see. All the picture is too dark and the lines (e.g. contour lines, additional contour line, etc.) are hard to see. Moreover, there are some useless particulars that should be deleted, for example I can see the number 47 and 55 without any sense. Please make this figure more readable.

Tables

Table 1. Please insert the unit of measurement of “abundance, 103”. Cell L-1?

References

L365. Please write “Eunotia“ in italics.

Author Response

Dear Reviewer,

Thank you for your comments and notes to our ms. We improve the text, tables and figures in respect of your comments. Responses to your comments are given below point by point.

General comments

The manuscript investigates the species richness and abundance of algae in seven swamps contaminated with oil products in the Khanty-Mansiysk Autonomous Okrug – Yugra, Russia. The study aimed to assess the oil pollution impact on this ecosystem by methods of algal bioindication.

My main concern regards that nutrients are not sufficiently considered. Species richness and abundance of algae are controlled by several environmental factors, and nutrients are among the most important ones. In this study there is no information regarding nutrients, neither from the literature or from the study itself (as no nutrient analyses was performed).

Response: added to the text and references:

In Results:

According to our studies (2014-2017), the concentration of nitrates in the soils of the Ershov oil field varied from less than 1 to 3.9 mg kg-1, and phosphorus from less than 5 to 190 mg kg-1 at individual research points. The surface water of the deposit contains insignificant concentrations of nitrates from 0.02 to 2.1 mg L-1 (at some points in some years), an average of 1.2 mg L-1. The phosphate content in surface waters varies significantly and reaches high values ​​(up to 5.6 mg L-1), with an average value of 0.3 mg L-1.

In Discussion:

The surface water of the Ershov deposit contains insignificant concentrations of nitrates from 0.02 to 2.1 mg L-1 (at some points in some years), an average of 1.2 mg L-1. It is known [23] that autotrophs in the swamps and wetlands have important role in the process of denitrification. The surface water of the Ershov bogs contains insignificant concentrations of nitrates from 0.02 to 2.1 mg L-1, an average of 1.2 mg L-1 that is in the range of nitrates in the water of the oligotrophic swamps such as Everglades (0.1 mg L-1  [24] and Poland swamps in Europe (3.0 mg L-1 ) [25].

In my opninion, the manuscript is generally nicely written and clear, but before the pubblication the nutrient availability of the study area should be clarifyied, maybe just showing literature data about it (if available). Some further changes are given below.

Response: added to the text and references the same response as above.

Specific comments

L77. Here only the site 1 is referred as control (background), but in figure 1 there are showed several background areas. What are those background areas? Other controls sites for those close to them? Please clarify it.

Response: corrected, the adequate map inserted

L157. Please change “thousand cells per 1 liter of water” to “x 103 cell L-1”.

Response: corrected

L158, 162, 163, 167, 168, 174, 175, 178, 179, 182 and 188. Change “thousand” to “x 103”.

Response: corrected

Figures

Figure 1. This figure is not easy to see. All the picture is too dark and the lines (e.g. contour lines, additional contour line, etc.) are hard to see. Moreover, there are some useless particulars that should be deleted, for example I can see the number 47 and 55 without any sense. Please make this figure more readable.

Response: corrected, new map inserted

Tables

Table 1. Please insert the unit of measurement of “abundance, 103”. Cell L-1?

Response: corrected

References

L365. Please write “Eunotia“ in italics.

Response: corrected

Reviewer 2 Report

Dear authors,

Your work is of interest to the scientific community. However, the article needs corrections. Please highlight the corrections made in the text with a half-time submission of the corrected manuscript. All comments in the attached file, as well as in the text of the article.

Author Response

Dear Reviewer,

Thank you for your comments and notes to our ms. We improve the text, tables and figures in respect of your comments. Responses to your comments are given below point by point.

Figure 1. The enlarged parts of Figure 1 (right) are not of good enough quality. It’s hard to distinguish anything. It is not clear what additional information the authors wanted to present on the right side of Figure 1.

Response: new figure added

Line 17 Bioindication revealed the salinity influence in the oil-contaminated sites. – What was the effect of salinity?

Response: yes, salinity, which come from the ground waters an as a results of technological process affected algae community algal species richness was decreased, we revealed some influence that discussed in the article, but this relationships cannot be discussed in Abstract, which limited.

Lines 18-19 The comparative floristic analysis shows the similarity of communities in sites surround the contaminated area in which the ecosystems were most rehabilitated. – This is not a clear proposal.

Response: corrected as: The comparative floristic analysis shows the similarity of communities in sites surround the contaminated area, the ecosystems of which have a long-term rehabilitation period.

Lines 85-86 «Varying degrees of water cut and a more or less dense projective cover of higher plants distinguish the studied areas». Need to expand this information a bit.

Response: corrected as: The studied areas differ in the degree of water cut and density of the projective cover of higher plants.

Lines 93-96 Field studies for the initial assessment - impact screening, were carried out in mid-July 2018, at 7 sites of the Ershov oil field, in the floodplain of the estuary of the Vakh river, in the Vakhovsky woodland area of the first floodplain terrace [7]. Why is the link here?

Response: corrected and moved to MM.

The studies were conducted on the territory of the KMAO-Yugra oil production area, in the floodplain of the Vakh River, in the Vakhovsky woodland area of the first floodplain terrace [7]. The water of the Vakh River basin, to which the studied areas belong, are weakly mineralized, bicarbonate.

Lines 104-105 «The mineral oil concentration was assessed in the Nizhnevartovsk State University chemical laboratory as mg of mineral oil per kg of the bog mass surface». How were samples taken (water or?) to determine the oil content. It is necessary to add a methodology for the selection and determination of oil content and a link to the methodology.

Response: added as:

The mineral oil concentration was assessed in the Nizhnevartovsk State University chemical laboratory. Samples of the swamp mass (soil) were taken using the envelope method from a depth of 0–25 cm. The size of the test site was 10 × 10 m. A combined sample was made, weighing 1 kg, by mixing five point samples of 200 g each, which was placed in a plastic bag and numbered. Soil samples were taken in bags made of polyethylene.

Samples of the swamp mass (soil) were dried at room temperature to an air-dry state. Then, mechanical inclusions (undecomposed roots, plant residues, stones, etc.) were removed, crushed using a laboratory homogenizer, and rubbed through a sieve with a mesh diameter of 0.5 mm. A sample of bog mass (soil) weighing 100 g was taken from the sample, which is dried in air to constant weight. For analysis, two parallel samples of 5 g each were used. The content of oil products was determined by IR spectrometry on analyzers of oil products. The method consists in the extraction of oil products from the swamp mass (soil) and bottom sediments with carbon tetrachloride, the chromatographic separation of oil products from related organic compounds of other classes, and the quantitative determination of oil products by the absorption intensity in the infrared region of the spectrum.

Line 119 the bog mass surface. Oil content in the water? It is necessary to add explanations to the methodology.

Response: added text as the response above

Table 1. Chemical and biological variables in the studied sites of the sphagnum bogs of the Ershov oil field in the Vakh River with geographical coordinates and year of impact. In my opinion, in Table 1 it is better to leave only Chemical variables and change the name accordingly. Quantitative indicators of biological communities are usually described after the species composition of the communities.

Response: corrected the Table name as “Averaged chemical variables in the studied sites of the sphagnum bogs of the Ershov oil field in the Vakh River with geographical coordinates, year of impact, and rehabilitation state” and abundance data removed.

Line 129 This diversity is rather rich because early studied surround algal habitat have similar species richness [8, 11-14]. This proposal is not about diversity (usually measured by the Shannon index), but about species richness.

Response: corrected as: This species richness is rather rich because early studied surround algal habitat have similar species richness [8, 11-14].

Table 3. «Calculation results of taxonomic and bioindication analysis of algae and cyanobacteria composition in the studied sites of the bogs of the Ershov oil field in the Vakh River of the KMAO-148 Yugra». The table also provides data on the abundance, which should be reflected in the name of the table. Besides, algae abundance data are given in Table 1. How are they related or not related? Also, the algae abundance data in Tables 1 and 3 are different.

Response: corrected, abundance removed from Table 1

It is best to divide Table 3 into two. One can provide information on the species richness of departments and data on numbers. In the second - the results of bioindication.

Response: This approach can multiply the number of tables, but we try to concentrate the information as much as possible. In any case, it can be divided after recommendation of technical editor.

Lines 154-187. When describing Study Sites, the information shown in Table 1 is repeated. Besides, some data does not match in the table, and the text does not match. In my opinion, the information in the text is better to give in a comparative aspect, without repeating the data in table 1

Response: name and information in Table 1 and text corrected

Figure 3a, b. - need to be moved to the description of Study Sites.

It is necessary to add the site numbers in all figures for greater clarity and understanding of the differences between the distributions of the analyzed indicators in the surveyed areas.

It is necessary to improve the resolution of drawings (quality).

Response: These maps is constructed with a new approach of data analysis and therefore represented the work results. So, we cannot moved it to the Study site description. The number of study site are added to Figure 3a.

Lines 222-227. The conclusion about the relationship of algae with salinity is not correct, since: 1. Data on salinity are not available. 2. Euryhaline (mesogalobic) species are a priori more resistant to pollution, compared with halophobic species. Therefore, the identified distribution of algae species cannot be associated with the influence of salinity.

Response: Data on salinity added to the text as averaged only because we cannot construct map without data in each site. The chlorides concentration was not identified in current study. That because we revealed salinity influence with bioindication. In this case, both mentioned ecological categories are sensitive, but to different range of salinity: mesohalobes can survive in high chlorides concentration whereas halophobic species avoid it. Therefore, the data about species of each mentioned group indicated the range of salinity that they prefer. It allow as to reveal the areas that have most saline waters.

Lines 282-283. «Figure 9 shows that environmental variables divided into three different groups». But the figure shows that external factors formed only two groups - one with pH, the other with saprobity index and oil content. A clearer description of Figure 9 is required.

Response: corrected as:

Relationships between biological (number of species in taxonomic division from Table 3) and chemical variables (from Table 1) of studied sites in the Ershov oil field was calculated in CANOCO program. Figure 9 shows that environmental variables divided into two different groups. Therefore, mineral oil and organic pollution as Index S were impacted algal community mostly in site 7 with euglenoid species preferring as a result (red dashed line). Water pH was only one variable that affected site 2 community with cyanobacteria flourishing (blue dashed line). Such biological variables as species richness and abundance that can take into account as variables, high level of which characterize environment most comfortable for algae development were represented the third group in Figure 9 (black dashed line). Sites 2 and 5 communities were most abundant and species-rich with preferences for diatom and Xanthophyceae algae. Therefore, these sites can be assessed as most rehabilitated and close to the control site 1.

Lines 316-317 “Comparative floristic analysis show the similarity of communities in sites surround the contaminated area and in which the ecosystems were most rehabilitated”. What do you mean?

Response: corrected as:

Comparative floristic analysis show the similarity of communities in sites surround the contaminated area and in which the ecosystems stay under rehabilitation of 10-20 years.

In general, it is also necessary to discus:

what environmental indicators are the background for the surveyed region - pH, oil content in soils (?), temperature, mineralization. How different are the environmental indicators of the background plot (number 1) from the background indicators in the region ?.

Response: added to the text in Discussion part, as in response below.

-You also need to compare the composition of the algal communities of the examined swamps with the swamp communities of the same region located far from anthropogenic impact. Is there any published data? What is the background composition of algae for the swamps of the surveyed region? What species dominate?

Response: added to the text in Discussion part, as in response below.

-Since some discussion of the results is given in the text itself, it is generally difficult to conclude whether the authors compared their results with existing data. It would be better and clearer if the discussion authors were allocated in a separate section.

Response: the Discussion part is formed as:

The surface water of the Ershov deposit contains insignificant concentrations of nitrates from 0.02 to 2.1 mg L-1 (at some points in some years), an average of 1.2 mg L-1. It is known [23] that autotrophs in the swamps and wetlands have important role in the process of denitrification. The surface water of the Ershov bogs contains insignificant concentrations of nitrates from 0.02 to 2.1 mg L-1, an average of 1.2 mg L-1 that is in the range of nitrates in the water of the oligotrophic swamps such as Everglades (0.1 mg L-1  [24] and Poland swamps in Europe (3.0 mg L-1 ) [25].

The Ershov oil deposit bogs algae species list included 91 taxa, mostly of which are affiliated to four taxonomic Divisions: Charophyta, Bacillariophyta, Cyanobacteria, and Euglenophyta. This species richness is rather rich in comparison to studied algal habitats surround. In the zone of sphagnum bogs in surrounded areas where direct pollution is not observed, the species composition of the identified algae in the bogs of the Agan and Novo-Pokursky deposits, respectively, is 76 and 109 species and intraspecific taxa (according to samples September 2018). So, in the bogs of the Agan deposits community were dominated by species of diatom, euglenoid, and green algae: Coenococcus planctonicus (724.2 x 103 cells L-1), Trachelomonas planctoica (366.6 x 103 cells L-1), and Rhopalodia gibba (228.2 x 103 cells L-1). In the Novo-Pokursky deposits algae community were found domination of diatoms and cyabobactera species: Rhopalodia gibba (765.5 x 103 cells L-1), Aphanothece clathrata (630.2 x 103 cells L-1), and Microcystis grevillei (352.8 x 103 cells L-1) [26]. In the Ershov bogs referenced site the Charophyta algae were dominated: Closterium baillyanum (30.3 x 103 cells L-1) and Actinotaenium cucurbitinum (1.8 x 103 cells L-1). That is incomparable low than the abundance of dominants in surrounded bogs. In polluted sites the algal abundance decreased but the taxonomic spectrum of dominants increase with addition of diatom, cyanobacteria, and Chrysophyta species.

The important process for the wetlands and swamps is eutrophication that defined as increased availability of elements that limit primary production and is usually associated with surface water pollution in swamps [27]. The organic pollution in the Ershov bogs is expressed in the Indices of saprobity S, which varied in the range of Class 2-3 of Water Quality [10]. Therefore, we can to conclude on the base of nutrient concentration and Index S that the Ershov bogs still in the oligotrophic state even after the oil pollution impact.

The bioindicator analysis was implemented with the statistically generated maps in the first time for the bog surface but not only for the waterbody surface area. Distribution of ecological groups and environmental variables with the maps comparison approach, which we early used for the water surface and for the catchment basin analysis [18], help us to reveal the relationships between oil concentration in the bog surface mass and distribution of salinity and organic pollution variables. Now we have not comparable materials from the surrounded area bogs because this approach was used in first time but the correlation of salinity and nutrients impact is known for the bogs and swamps from different regions [27].

Round 2

Reviewer 1 Report

Authors have answered almost all my concerns.

Authors added information about nutrients, but they still need to make clear two points about that:

How did they analyze those nutrients? How did they collect samples for those analyses? Please add a paragraph in Materials and methods about the nutrient analyses. Did you find any significant differences among the different sites about the nutrient concentrations?

Moreover, as in this new version authors added a distinct Discussion section, they should:

L 146. Rename as “4. Results” Avoid discussing their results in the Results’ section, i.e. delete L206-207 “This species richness is rather rich because early studied surround algal habitat have similar species richness [8, 11-14”; L312-316. “We cannot compare the Divisional distribution in the disturbance (…) abundance and biomass of soil algae”; L358-359 “It let us allow that the oil production (…) as has been mentioned in the nearby area [15]”. Move them in the Discussion section. Rename as “5. Discussions”. 477-478. “The surface water of the Ershov deposit contains insignificant concentrations of nitrates from 0.02 to 2.1 mg L-1 (at some points in some years), an average of 1.2 mg L-1.” Remove this part as you will repeat that in the subsequent lines. Rename as “6. Conclusions”

Author Response

Dear Editor,

Thank you and the Reviewer for your comments. Please find below the responses to the Reviewer 1 comments point by point. Changes in ms were made with respect to your comments with the track change. Few corrections have been done in English editing.

With best regards,

Prof Sophia Barinova, Prof Olga Scorobogatova

Responses to comments of the Reviewer 1

Authors have answered almost all my concerns.

Authors added information about nutrients, but they still need to make clear two points about that:

How did they analyze those nutrients? How did they collect samples for those analyses? Please add a paragraph in Materials and methods about the nutrient analyses. Did you find any significant differences among the different sites about the nutrient concentrations?

Response:

Added reference for analysis: Alekin, O.A. Fundamentals of hydrochemistry. Leningrad: Gidrometeoizdat, 1953, 296 p.

In any case, we cannot compare the concentrations of nitrates between sites because it comes from different years on the different stations, mentioned in the text.

Moreover, as in this new version authors added a distinct Discussion section, they should:

L 146. Rename as “4. Results”

Response: renamed

Avoid discussing their results in the Results’ section, i.e. delete L206-207 “This species richness is rather rich because early studied surround algal habitat have similar species richness [8, 11-14”;

L312-316. “We cannot compare the Divisional distribution in the disturbance (…) abundance and biomass of soil algae”;

L358-359 “It let us allow that the oil production (…) as has been mentioned in the nearby area [15]”. Move them in the Discussion section.

Response: moved

Rename as “5. Discussions”.

Response: done

477-478. “The surface water of the Ershov deposit contains insignificant concentrations of nitrates from 0.02 to 2.1 mg L-1 (at some points in some years), an average of 1.2 mg L-1.” Remove this part as you will repeat that in the subsequent lines.

Response: removed

Rename as “6. Conclusions” 

Response: done

Reviewer 2 Report

Dear authors, thanks for the corrections. There are still small notes that are easy to correct.

Lines 143-144. The summer season was chosen because the impact, water level, development of algal communities in this period were maximal. What is the impact (of pollution?)? Was the water level maximal too?

Lines 153-166. Please, add the references for the methods that were used.

Results and Discussion

Lines 206-207. “This species richness is rather rich because early studied surround algal habitat have similar species richness”. Sorry, but this sentence remains unclear. Why is such a conclusion? Have previous studies been larger? Were the same sites studied? Please clarify what you mean.

Lines 312-316. We cannot compare the Divisional distribution in the disturbance gradient in close related research because usually an only total list of algae has been represented [8,314 11-14, 15]. But our results were similar to the experimental research where oil pollution caused an algal community to be rearranged [22], with a change in species composition, a decrease in the cells abundance and biomass of soil algae.

Discusions.

Since the discussion is highlighted in a separate section, please delete these words from the title of subsection 4 “Results and Discussion». It was better to transfer the information – lines 206-207, 312-316 - to the discussion section.

Author Response

Dear Editor,

Thank you and the Reviewer for your comments. Please find below the responses to the Reviewer 2 comments point by point. Changes in ms were made with respect to your comments with the track change. Few corrections has been done in English editing.

With best regards,

Prof Sophia Barinova, Prof Olga Scorobogatova

Responses to comments of the Reviewer 2

Dear authors, thanks for the corrections. There are still small notes that are easy to correct.

Lines 143-144. The summer season was chosen because the impact, water level, development of algal communities in this period were maximal. What is the impact (of pollution?)? Was the water level maximal too?

Response: as mentioned, water level was maximal too. Corrected as:  The summer season for the sampling was chosen because the impact on the aquatic ecosystem of the bogs is maximal in the summer ice-open period when the bog water level and development of algal communities is also maximal.

Lines 153-166. Please, add the references for the methods that were used.

Response: corrected, replaced dominance to revealed

Results and Discussion

Lines 206-207. “This species richness is rather rich because early studied surround algal habitat have similar species richness”. Sorry, but this sentence remains unclear. Why is such a conclusion? Have previous studies been larger? Were the same sites studied? Please clarify what you mean.

Response: Rephrased, added the references

Lines 312-316. We cannot compare the Divisional distribution in the disturbance gradient in close related research because usually an only total list of algae has been represented [8,314 11-14, 15]. But our results were similar to the experimental research where oil pollution caused an algal community to be rearranged [22], with a change in species composition, a decrease in the cells abundance and biomass of soil algae.

Response: Rephrased

Discusions.

Since the discussion is highlighted in a separate section, please delete these words from the title of subsection 4 “Results and Discussion». It was better to transfer the information – lines 206-207, 312-316 - to the discussion section.

Response: done
